# Protective Effect of *Antrodia cinnamomea* Extract against Irradiation-Induced Acute Hepatitis

**DOI:** 10.3390/ijms20040846

**Published:** 2019-02-15

**Authors:** Tsu-Hsiang Kuo, Yueh-Hsiung Kuo, Chun-Yu Cho, Chih-Jung Yao, Gi-Ming Lai, Shuang-En Chuang

**Affiliations:** 1Graduate Institute of Life Sciences, National Defense Medical Center, Taipei 11490, Taiwan; dennissylvie@gmail.com; 2National Institute of Cancer Research, National Health Research Institutes, Miaoli 35053, Taiwan; chunyu0116@nhri.org.tw; 3Department of Chinese Pharmaceutical Sciences and Chinese Medicine Resources, China Medical University, Taichung 40402, Taiwan; kuoyh@mail.cmu.edu.tw; 4Department of Biotechnology, Asia University, Taichung 41354, Taiwan; 5Cancer Center, Wan Fang Hospital, Taipei Medical University, Taipei 11696, Taiwan; yao0928@tmu.edu.tw (C.-J.Y.); gminlai@tmu.edu.tw (G.-M.L.); 6Department of Internal Medicine, School of Medicine, College of Medicine, Taipei Medical University, Taipei 11031, Taiwan

**Keywords:** *Antrodia cinnamomea*, radiation-induced liver disease, radiation-induced liver disease (RILD), reactive oxygen species, ROS, acute hepatitis

## Abstract

Radiotherapy for treatment of hepatocellular carcinoma causes severe side effects, including acute hepatitis and chronic fibrosis. Complementary and alternative medicine (CAM) has emerged as an important part of integrative medicine in the management of diseases. *Antrodia cinnamomea* (AC), a valuable medicinal fungus originally found only in Taiwan, has been shown to possess anti-oxidation, vaso-relaxtation, anti-inflammation, anti-hepatitis, and anti-cancer effects. In this paper we evaluate the protective effects of ethanol extract of *Antrodia cinnamomea* (ACE) against radiotoxicity both in normal liver cell line CL48 and in tumor-bearing mice. In CL48, ACE protects cells by eliminating irradiation-induced reactive oxygen species (ROS) through the induction of Nrf2 and the downstream redox system enzymes. The protective effect of ACE was also demonstrated in tumor-bearing mice by alleviating irradiation-induced acute hepatitis. ACE could also protect mice from CCl_4_-induced hepatitis. Since both radiation and CCl_4_ cause free radicals, these results indicate that ACE likely contains active components that protect normal liver cells from free radical attack and can potentially benefit hepatocellular carcinoma (HCC) patients during radiotherapy.

## 1. Introduction

Radiation therapy remains an important regimen for the treatment of unresectable hepatocellular carcinoma (HCC). While radiotherapy is generally much more location specific and theoretically has fewer side effects than systemic chemotherapy, irradiation-induced hepatitis is still difficult to avoid. One of the important liver injuries due to irradiation is radiation-induced liver disease (RILD), or radiation hepatitis [1]. The clinical pathophysiology of RILD is characterized by hepatomegaly, abdomen ascites and elevation of aspartate transaminase (AST), alanine transaminase (ALT), and other liver enzymes [2]. However, the most effective measure against radiotherapy-induced hepatitis is the administration of preventive measures before radiotherapy; but there is currently no effective treatment available for RILD.

The primary effects of cellular exposure to ionizing radiation are direct destruction of DNA, lipids, and proteins, and indirect damage caused by the increased generation of reactive oxygen species (ROS) [3]. Oxidative DNA damage is responsible for the induction of many malignancies [4]. Chronic oxidative stress may induce cell apoptosis, progression of degenerative diseases, and irradiation-induced tissue injury [5]. Increasing evidence also demonstrates that the radiation-induced effects may also be mediated through disruption of redox homeostasis and intercellular communications [6]. In normal conditions, intracellular ROS are produced as essential signaling messengers that regulate cellular processes [7]. However, excess ROS produced by ionizing radiation are cytotoxic. To maintain cellular homeostasis, cells must regulate cellular redox levels [8]. Radiation-mediated cleavage of water molecules can generate massive amounts of hydrogen peroxide and damage the mitochondrial respiratory chain to further produce excessive superoxide radicals [9].

*Antrodia cinnamomea* (AC) (Chang & Chou) [10] is a valuable fungus originally found only in Taiwan. Like many other medicinal mushrooms, AC exhibits anti-oxidative and anti-inflammatory effects [11,12] and is used to treat liver diseases, hypertension, diarrhea [13], and cancers [14]. Crude extracts and active fractions of AC have been shown to ameliorate liver disorders caused by excessive alcohol consumption or chemical damage [15]. Recent studies have revealed the hepato-protective effects of the crude extracts or single constituents from AC [16,17]. A biological and phytochemical investigation of AC has also revealed their potent cytotoxic, anti-inflammatory, and hepato-protective activities of different constituents and pharmacologically active secondary metabolites [18]. The probable molecular mechanisms underlying the hepatoprotective effects of AC against damages caused by toxins, chemicals, and alcohol depend on the regulation of the cellular redox enzyme system [19,20]. In irradiation-induced tissue damage, AC is capable of inhibiting immuno-regulatory signaling to avoid the cytotoxic and inflammatory responses of immune cells in vitro [21]. However, there have been no reports published to investigate whether AC, or its active components, can alleviate the liver damage caused by irradiation.

Nuclear factor erythroid-2-related factor (Nrf2) is bound to Keap1 (the Kelch-like ECH-associated protein 1, a cytosolic protein). When levels of ROS increase, the conformationally changed Nrf2 separates from Keap1, and Nrf2 translocates to the nucleus to transactivate genes of antioxidant and cytoprotector proteins through binding to the antioxidant response element (ARE) of the gene promoters [22]. During liver injuries, oxidative stress and inflammation resulting from unregulated production of ROS are important events. Since Nrf2 is the master regulator of cellular redox system, it possesses potent anti-inflammatory and cytoprotective properties in liver cells [23].

Transactivation of redox genes by Nrf2 provides a host defense mechanism against oxidative injury and also contributes to the anti-inflammatory activity of cells and tissues. In the present study, we evaluated the hepatoprotective effects of the ethanol extract of fruiting bodies of AC against irradiation both in vitro and in animal models. We also observed that AC could also protect mice from CCl_4_-induced acute hepatitis. These results indicate that AC extract likely contains active components that protect normal liver cells from free radical attack through. This result is possibly due to up-regulation of Nrf2 and its downstream redox enzymes to eliminate free radicals.

## 2. Results

### 2.1. Radio-Protective Effects of Antrodia cinnamomea Extracts in Liver Cells

We first examined the cytotoxicity of various dosages (0, 50, 100, 150, and 250 Gy) of irradiation in the human liver cell line CL48 (Figure 1A). The effect of *Antrodia cinnamomea* extract (ACE) treatment (serial dilution from 250 μg/mL) on cell survival was examined by 3-(4,5-dimethylthiazol-2-yl)-2,5-diphenyltetrazolium bromide (MTT) assay (Figure 1B). As shown, growth of CL48 cells was inhibited by irradiation (Figure 1A) or ACE (Figure 1B) in a dose-dependent manner. At sub-lethal doses of 20, 40, and 80 μg/mL, ACE was able to protect CL48 cells from irradiation in the presence of three different dosages followed by MTT assay for cell survival after 48 h (Figure 1C). The Quality control of ACE was performed by thin layer chromatography and HPLC analysis (Appendix A).

### 2.2. ACE Protects Liver Cells from Irradiation-Induced Apoptosis

We examined the protective effects of ACE on irradiation-induced apoptosis of CL48 liver cells. The irradiation-induced caspase-3, -8, and -9 activities (Figure 2A–C) were attenuated by ACE treatment in a dose-dependent manner (Figure 2D). Irradiation-induced apoptosis, as examined by Annexin V and Propidium Iodide (PI) double staining followed by flow cytometry assay (Figure 2E). Percentages of the early and late apoptotic cells were combined and compared statistically. As shown, apoptosis was significantly inhibited by pre-treatment with ACE in a dose-dependent manner and decreased from 52% to 34% and 14% by ACE 40 μg/mL and 80 μg/mL, respectively (Figure 2F).

### 2.3. Evaluation of Direct Free Radical-Scavenging Activity of ACE in Solution

To determine if ACE has direct antioxidant activity, we assessed the free radical scavenging ability of ACE by using the stable 1,1-diphenyl-2-picrylhydrazyl (DPPH) free radical assay [24] and electron spin resonance (ESR) spectrometry [25]. As a result, ACE was found to possess only mild free radical-scavenging activity in vitro as illustrated by both DPPH assay (Figure 3B) and ESR assay (Figure 3C,D). As a positive control, NAC (*N*-acetyl-l-cysteine) was shown to possess strong free radical-scavenging activity in vitro in a dose-dependent manner (Figure 3A,C,D).

### 2.4. Irradiation-Induced ROS Scavenging Activity of ACE in Liver Cells

The cell-based DCFH-DA (2′,7′-dichlorofluorescin diacetate) assay was used to evaluate the free radical scavenging activity of ACE. Results showed that ACE could lower the irradiation-induced ROS levels (Figure 4B) in a dose-dependent manner and decreased 50% and 78% by ACE 40 μg/mL and 80 μg/mL, respectively (Figure 4G,H). NAC was used as a positive control (Figure 4D).

### 2.5. ACE Enhances the Irradiation-Induced Redox Enzymes Activities of Hepatocytes

Expression and activity levels of several important redox enzymes were evaluated in liver cells with or without ACE treatment. As a result, ACE was found to enhance the irradiation-induced expression of MnSOD and GPx in a dose-dependent manner both at the mRNA (Figure 5A) and protein (Figure 5B) and activity (Figure 5C,D) levels.

### 2.6. ACE Enhances the Expression and Nuclear Translocation of Nrf2 in Irradiated Hepatocytes

Expression of the upstream regulator of MnSOD and glutathione peroxidase (GPx), i.e., the redox-sensitive transcription factor Nrf2 (Nuclear factor-erythroid 2-related factor), in the nucleus vs. cytoplasm was examined by Western blot and immunofluorescence staining. As a result, it is evident that irradiation could induce nuclear translocation of Nrf2 and ACE may further increase its nuclear expression in a dose-dependent manner (Figure 6A). The results were also demonstrated by immunofluorescence staining for Nrf2 in CL48 cells irradiated with or without ACE. While irradiation alone could increase both nuclear and cytoplasmic expression of Nrf2 (Figure 6B, second panel), pre-treatment with ACE dramatically further enhanced this effect in a dose-dependent manner (Figure 6B, third and fourth panels). The Figure 6B quantity ratio of nucleus Nrf2 is expressed in Figure 6C.

### 2.7. ACE Alleviates the Irradiation-Induced Liver Inflammation In Vivo

AC has been demonstrated to be capable of alleviating liver inflammation in various animal models [26,27,28]. First, we re-confirmed the hepato-protective effect of ACE in CCl_4_-induced acute hepatitis mice (Appendix A). Here, we further demonstrate the hepato-protective effect of ACE in the irradiation-induced hepatitis mouse model. In the scheme of the animal model experiment, mice were divided into four groups, and each group had six mice (Figure 7A). As a result, both ACE doses were found to reduce the hepatitis markers Aspartate aminotransferase (AST) (Figure 7B) and Alanine aminotransferase (ALT) (Figure 7C). The hepato-protective effect of ACE on irradiation-induced hepatitis was also demonstrated histologically by haematoxylin and eosin (H&E) staining in decreasing hepatitis area of the liver samples sections (Figure 7D). As demonstrated by immunohistochemical staining, the expression and nuclear translocation of Nrf2 were also significantly enhanced and nuclear localized in the mice of ACE treatment groups (Figure 7E). The Figure 7E quantity ratio of nucleus Nrf2 is expressed in Figure 7F. 

## 3. Discussion

Surgical resection and liver transplantation are the primary treatment modalities for hepatocellular carcinoma (HCC). Various radiotherapy modalities are suitable for treating unresectable HCC and could be used as a post-surgery adjuvant therapy or in concomitant radio/chemotherapy. Radiation-induced liver disease (RILD) is one of the major limits of HCC radiotherapy and may compromise patients’ life quality and, due to various clinical conditions, only 30% to 40% of patients may benefit from radiotherapy [29]. Along with the development of three-dimensional conformal radiotherapy, image-guided high-dose radiation treatment could be delivered to focal liver tumor areas. Several studies have demonstrated that stereotactic body radiotherapy (SBRT) with 24 to 60 Gy administered in three to six fractions irradiated from different directions led one-year overall survival rates to rise 48% to 82% [30,31]. Since the tumor receives a much higher dose than adjacent non-tumor tissues, RILD is significantly reduced in SBRT. However, RILD still result in a general decline in liver function and markedly elevated transaminases and/or jaundice due to accumulating radiotoxicity in surrounding non-tumor tissues, either in an acute response within a few days or as a late response months to years after radiotherapy [32]. According to current radiotherapy administered in the form of SBRT, we believe that, with optimal doses of ACE, RILD can be significantly reduced without compromising the efficacy of HCC radiotherapy. In our current tumor-bearing mouse model study, neither the radiation doses nor the ACE doses used affected tumor growth in 25 days of our experimental period.

A few compounds, e.g., amifostine and α-tocopherol (vitamin E), have been shown in animal-model studies to reduce radiotoxicity by reducing lipid peroxidation and maintaining endogenous antioxidant defense mechanisms [33]. ACEs have been reported to possess hepato-protective effects against chemical-induced liver damage [15,28]. Several constituents of AC fruiting bodies and mycelia have been demonstrated to possess anti-inflammatory or hepato-protective effects [34,35,36]. Furthermore, some phytochemicals are capable of eliciting Nrf2 activation in the liver or HCC cells. For example, pomegranate polyphenolic constituents could induce hepatic antioxidants through dissociating the Nrf2-Keap1 complex in facilitating the Nrf2 nuclear translocation and activation [37]. In the current study we have demonstrated the protective effects of ACE against irradiation-induced hepatitis both in vitro and in vivo. Consistent with the literature [38], apoptosis induced by irradiation was mediated through the intrinsic pathway, i.e., activation of caspase-3 and caspase-9 but not caspase-8 (Figure 2D). Our results conclusively demonstrate that ACE significantly reduced the irradiation-induced apoptosis in a dose-dependent manner in CL48 cells. In elucidating the mechanisms of ACE’s protective effects, we demonstrated that several redox enzymes were up-regulated by ACE treatment, probably through activation of the transcription factor Nrf2 in liver cells. Increased amounts of ALT and AST were observed in radiation-induced acute hepatitis in mice. In vivo study also showed that pretreatment with ACE significantly prevented AST and ALT elevation in the serum of mice.

Only a few studies have reported the effects of ACE or its pure components on up-regulating Nrf2 expression to reduce lipopolysaccharide (LPS)-induced oxidative stress or inflammation and ethanol-induced liver damage [20,39]. Recent evidence also indicates that AC may induce Nrf2 activation and attenuate the LPS-induced inflammatory response via dissociation of Nrf2 from Keap1 in macrophages [40]. While RILD pathogenesis has been reported to be associated with vascular changes, increased activation of growth factors and cytokines, and collagen synthesis [41], the molecular mechanisms underlying the pathogenicity of RILD remain unclear. In the long term, RILD patients may suffer from liver fibrosis; and expression of the key growth factors responsible for liver damage responses—such as tumor necrosis factor alpha (TNF-α), transforming growth factor beta (TGF-β), and hedgehog (Hh) signaling pathway proteins—could be significantly up-regulated [42]. There is also evidence that ACEs could alleviate TGF-β-induced early liver fibrosis caused by alcohol treatment [43] and protect liver from injury caused by TNF-α activation-mediated acetaminophen toxicity [35]. We postulate that ACEs possess the abilities to alleviate fibrogeneis in RILD patients and potentially may have synergistic anti-cancer effects when used in combination with irradiation therapy. 

On the other hand, AC has also been reported to possess anti-cancer activities. AC has been reported to have cytotoxicity and induce apoptosis in cancer cells [14,44]. The reasons underlying the AC’s simultaneous anti-cancer and cyto-protective effects may lie in its dosages or concentrations used [44]. Recent studies have demonstrated that ACEs could inhibit TNF-α-induced migration/invasion in human endothelial cells. There is also evidence demonstrating that ACEs improve therapy efficacy in cancer patients [45,46]. However, the ROS role in cancer therapy is still controversial, and redox regulation in tumor cell epithelial-mesenchymal transition is still being discussed [47]. How to enhance the sensitivities of cancer cells to radiotherapy or chemotherapy and alleviate the damage of normal cells are important issues of patient prognosis. Since ROS have effects on inflammation, tissue damage and regeneration, and cancer metastasis, their regulation of the redox enzyme system in cancer or normal cells may play a crucial role in therapy. 

## 4. Materials and Methods

### 4.1. Irradiation Treatment

Cells were seeded and grown to 80% confluency. The cells were irradiated with an RS2000 X-ray irradiator (Rad Source Technologies, Coral Springs, FL, USA). In animal studies, each of the 6 to 8-week-old mice was irradiated with a total dosage of 15 Gy (5 Gy every 24 h for three times) on the upper abdominal region to induce acute liver damage.

### 4.2. Antrodia Cinnamomea from Fruiting Bodies in Cultivation Bags

The AC fruiting body samples were kindly provided by Well Shine Biotechnology Development Co. (Taipei, Taiwan). The authentication of this wild fungal strain was performed by the Food Industry Research and Development Institute (Hsinchu, Taiwan) using DNA sequencing. To enable the growth of fruiting bodies, the fungus was first cultured in petri dishes then planted in cultivation bags. On harvest, the fruiting bodies were collected and soaked in ethanol for 30 days. The triterpenoid-rich ethanol extract was collected by centrifugation and completely dried by vacuum evaporation. The extracts were dried respectively and mixed at a fixed ratio; they contained 15% to 20% triterpenoids.

### 4.3. Preparation of Ethanol Extracts of Antrodia cinnamomea

As in our previous study [48], 5 g of the air-dried AC fine powders were shaken with 100 mL of 95% ethanol for 24 h at room temperature. The supernatant of the extraction was centrifuged at 2250× *g* for 30 min to remove the insoluble parts. The extracts were concentrated with a vacuum evaporator to nearly dry and were re-dissolved in 95% ethanol. The AC ethanol extracts (ACE) stock solution was stored at −20 °C for later use. The different batches of ACE quality control analysis results are shown in Appendix A.

### 4.4. Cell Culture and Viability Assay

Normal human embryonic liver cell line CL48 cells (American Type Culture Collection) were cultured in high glucose DMEM with 10% fetal bovine serum (FBS), 0.1 mg/mL of streptomycin, and 100 units/mL of penicillin, at 37 °C in a humidified atmosphere containing 5% CO_2_. In cell-counting proliferation assays, 2.5 × 10^5^ cells were seeded into each well (BD Falconì, 24-well plates). At harvest, cells were stained with trypan blue solution (Sigma Aldrich, St. Louis, MO) and counted with a hemacytometer. Cell viability was also evaluated by MTT assay as previously described [49]. In brief, CL48 cells (6 × 10^3^) were seeded into each well of a 96-well plate and cultured for various intervals. Freshly prepared MTT was incubated, at a concentration of 0.5 mg/mL, with cells for 4 h. The formazan crystals were dissolved in 100 μL of DMSO and the optical density was measured by an enzyme-linked immunosorbent assay (ELISA) reader (Bio-tek Instruments, Winooski, VT, USA) at 570 nm. Each experiment was performed independently at least three times, and each in triplicate.

### 4.5. Apoptosis Analysis

As previously described [50], activities of caspase-3, caspase-8, and caspase-9 were determined with colorimetric assay kits (BioVision, Milpitas, CA, USA). Cells were irradiated with or without a 16-hour pretreatment with *Antrodia cinnamomea* extracts. After incubation for 48 h, cells were pelleted, lysed, and kept on ice for 10 min. Supernatants were collected by centrifugation at 10,000× *g* at 4 °C for 3 min. DEVD-pNA (Asp-Glu-Val-Asp p-nitroaniline), IETD-pNA (Ile-Glu-Thr-Asp p-nitroaniline), and LEHD-pNA (Leu-Glu-His-Asp p-nitroaniline) were added to the supernatant, respectively, to measure the caspase-3, caspase-8, and caspase-9 activities. Each sample was further incubated at 37 °C for 1 h and the fluorescent signals were measured by an ELISA reader. 

We used Annexin V-FITC apoptosis detection kit (BD Biosciences, Franklin Lakes, NJ, USA) to detect early and late apoptosis. At harvest, cells were collected, washed, and resuspended in binding buffer (10 mM 4-(2-hydroxyethyl)-1-piperazineethanesulfonic acid (HEPES), 144 mM NaCl, 25 mM CaCl_2,_ pH7.4). Annexin V-FITC (0.2 μg/μL) and PI (0.05 μg/μL) were added and incubated in the dark for 20 min. Cells were then subjected to flowcytometric analysis.

### 4.6. Stable Free Radical Scavenging Activity Assay

Stable free radical scavenging activity was measured by DPPH (1, 2-diphenyl-2- picrylhydrazyl, a stable nitrogen-centered free radical) assay using ESR spectrometry. DPPH was dissolved in 100% ethanol at 100 μM. Each test material (100 μL) was added to 900 μL of the resulting dark-blue DPPH radical solution in a cuvette, and absorbance at 517 nm was measured. ESR spectrometry was mainly for the detection of free radicals in solutions or in in vitro systems containing subcellular components. Spectra were taken with a Varian E-9 X-band spectrometer (9.5 GHz) with a field modulation frequency of 100 KHz by using a microwave power of 10 mW and modulation amplitude of 0.4 G.

### 4.7. Measurement of Intracellular Free Radicals

The oxidation of 2′,7′-dichlorofluorescin diacetate (DCFH-DA) was used to measure the intracellular ROS levels. The DCFH diacetate crossed the cell membrane and was de-esterified to DCFH. The intracellular free radicals or ROS oxidized DCFH-DA to fluorescent DCFH, and the fluorescence intensity was measured. With or without pre-incubation with ACE or *N*-acetyl-l-cysteine (NAC) for 12 h, cells were irradiated at various dosages, collected 24 h later, and were subjected to free radical measurement assay. For positive control, cells were exposed to 700 μM H_2_O_2_ for 30 min before free radical measurement. The reaction took place with cells and 1 μM DCFH-DA in 2 mL PBS on ice in dark for 15 min and analyzed by flow cytometer. 

### 4.8. Measurement of Superoxide Dismutase and Glutathione Peroxidase Activities

Cellular SOD activity was measured by using Cayman’s Superoxide Dismutase Assay Kit (Cayman Chemical, Ann Arbor, MI, USA). The reaction product of formazan dye was monitored by measuring the absorbance at 440 to 460 nm using a plate reader. The final SOD activity was calculated using the bovine erythrocyte SOD standard activity calibration curve and expressed in U/mL. Cellular GPx activity was measured by using the Glutathione Peroxidase Assay Kit (Cayman Chemical, Ann Arbor, MI). The GPx activity was indirectly measured by coupled reaction with Glutathione disulfide (GSSG) (the oxidized form of glutathione) and glutathione reductase (GR). The oxidation of NADPH to NADP+ was accompanied by a decrease in absorbance at 340 nm. The final GPx activity was calculated and expressed in nmol/min/mL. Each experiment was repeated three times independently.

### 4.9. Semi-quantitative RT-PCR and Western Blot Analysis

Total RNA was extracted from cells by using reagent TRIzol (Invitrogen Life Technologies, Carlsbad, CA, USA). Cellular proteins were extracted by using RIPA lysis buffer (1 × PBS, 1% Nonidet P-40, 0.5% sodium deoxycholate, 0.1% SDS) with protease inhibitor cocktail (Pierce, Rockford, IL, USA). Equal amounts of proteins were separated by 10% SDS-polyacrylamide gel electrophoresis followed by Western blot analysis. For cellular cytoplasmic and nuclear protein analysis, cell lysates were prepared and separated according to the instructions of the NE-PER nuclear and cytoplasmic extraction kit (Thermo Fisher Scientific, Waltham, MA, USA)

### 4.10. Immunofluorecscence Staining of Nrf2

Cells were cultured on round coverslips in 6-well plate to 70% confluence, washed with PBS and immersed in 0.5% Triton-X100 to permeabilize the membranes, fixed with 4% paraformaldehyde (PFA) in PBS for 15 min at room temperature, and then blocked with BSA in PBST (PBS containing 0.5% Tween-20) at 4 °C overnight. Polyclonal anti-Nrf2 antibody (1:500) (sc-722, Santa Cruz Biotechnologies, Santa Cruz, CA, USA) and appropriate fluorescein-conjugated secondary antibodies were used to detect Nrf2. The samples were counter-stained with 4′,6-diamidino-2-phenylindole (DAPI) and mounted with ProLong Gold Antifade Mountant medium (Life Technologies, Carlsbad, CA). Fluorescence images were captured using the same exposure time with a Zeiss Objective EC Plan-Neofluar 40×/1.30 Oil DIC lens on a Zeiss AxioPlan 2 fluorescence microscope (Carl Zeiss AG, Oberkochen, BW, Germany) in conjunction with Leica CytoVision software (Leica, Wetzlar, Hesse, Germany).

### 4.11. Tumor Allografts and In Vitro Molecular Imaging

Orthotopic mouse liver tumors were established in Balb/c mice using the mouse BNL/Luc cell line. Female 5 to 6-week-old Balb/c mice were purchased from the National Laboratory Animal Center (Taipei, Taiwan). All mice were conventionally housed with water and food and kept at 25 °C in an air-conditioned environment under 12-h light/dark control. For orthotopic implantation of cancer cells, each experimental group contained six mice. ACE was administered by oral gavage. The mice were fixed under aseptic conditions during general anesthesia with isoflurane 2.5% (volume of atmosphere). The BNL/Luc cells (10^5^) were injected into one hepatic lobe of each Balb/c mouse. The mice were monitored by bioluminescence imaging using IVIS imaging system (Xenogen, Alameda, CA, USA) to check for tumor growth weekly. Mice received irradiation on day 13 after tumor allograft. Blood samples were collected by orbital sinus sampling every 24 h. At 48 h, three mice from each group were exposed to CO_2_ to achieve euthanasia, and the liver tissue samples were collected for immunohistochemical (IHC) staining (Nrf2) and H&E staining. At sacrifice, all the remaining mice were euthanized in their own cages by introduction of 100% CO_2_. All animal-model experiments were performed in accordance with a protocol approved by the Institutional Animal Care and Use Committee of the National Health Research Institutes (NHRI-IACUC-104070-A).

### 4.12. Immunohistochemical Staining

Liver tissue samples were obtained from three mice in each experimental group, respectively, after irradiation 48 h. Samples were sectioned (5 µm thickness) and fixed with 4% paraformaldehyde. The sections were stained with H&E and analyzed. IHC staining was performed on the paraffin-embedded sections. After de-paraffinization, the sections were blocked with 5% bovine serum albumin for 1 h at room temperature in PBS containing 0.1% Triton X-100. The sections were then incubated overnight at 4 °C with the polyclonal anti-Nrf2 antibodies (1:500) (sc-722, Santa Cruz Biotechnologies, Santa Cruz, CA, USA) and appropriate secondary antibodies. The sections were then gently rinsed three times with PBS followed by hybridization with the secondary antibody at room temperature in a humid chamber. After immunohistochemical staining and counterstaining with DAPI, the sections were examined under the microscope. 

### 4.13. Statistical Analysis

The results are expressed as mean ± SD (*n* ≥ 3) of at least 3 independent experiments, each experiment was done in triplicate. Data were analyzed using one-way analysis of variance (ANOVA), followed by post-hoc Tukey multiple comparison tests with significance level set at *p* < 0.05. Statistical analysis was performed using SPSS version 20.0 (SPSS Inc., Chicago, IL, USA).

## 5. Conclusions

Here we demonstrate that ACE was able to protect animals from irradiation-induced liver damage and hepatitis effectively by up-regulating the redox system during irradiation treatment. While ACE was able to lower the side effects of radiotherapy, it would be worthwhile to further explore the possibility that ACE may also affect, or even increase, the efficacy of radiotherapy. 

## Figures and Tables

**Figure 1 ijms-20-00846-f001:**
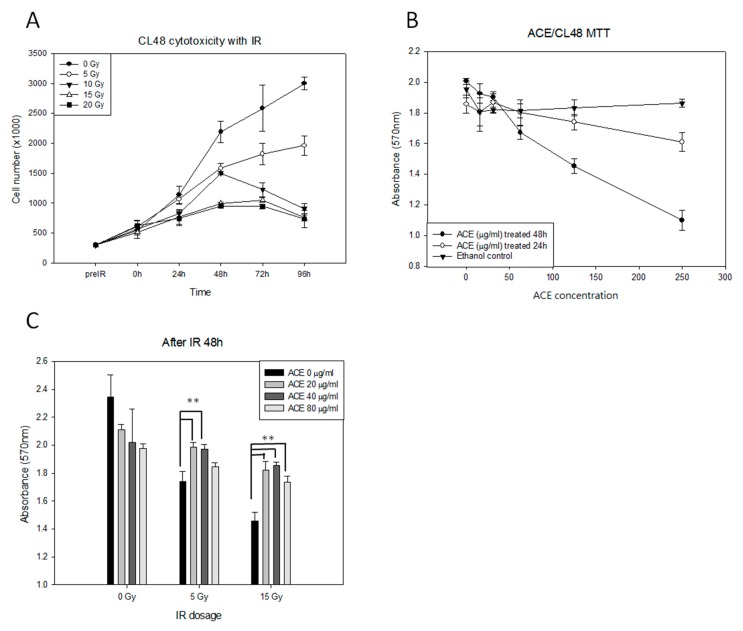
The protective effects of *Antrodia cinnamomea* extract (ACE) against irradiation in human normal liver CL48 cells. (**A**) Cytotoxicity of irradiation in CL48. Cells were treated with different dosages (0, 5, 10, 15, and 20 Gy) of irradiation and counted by hemocytometer at intervals as indicated. (**B**) Determination of the optimal ACE dosages. Cells were treated with different concentrations of ACE for 24 h or 48 h, followed by 3-(4,5-dimethylthiazol-2-yl)-2,5-diphenyltetrazolium bromide (MTT) assay for cell survival. For the control group, the same concentrations of ethanol were used for each ACE treatment experiment. (**C**) The protective effect of ACE against irradiation. Cells were pre-treated with different concentrations of ACE for 16 h followed by various dosages of irradiation; they were evaluated by MTT assay 48 h later. Cell survival was evaluated by MTT assay. Results were obtained from three independent experiments; each experiment was done in triplicate. **, *p* < 0.01, as compared with the control group.

**Figure 2 ijms-20-00846-f002:**
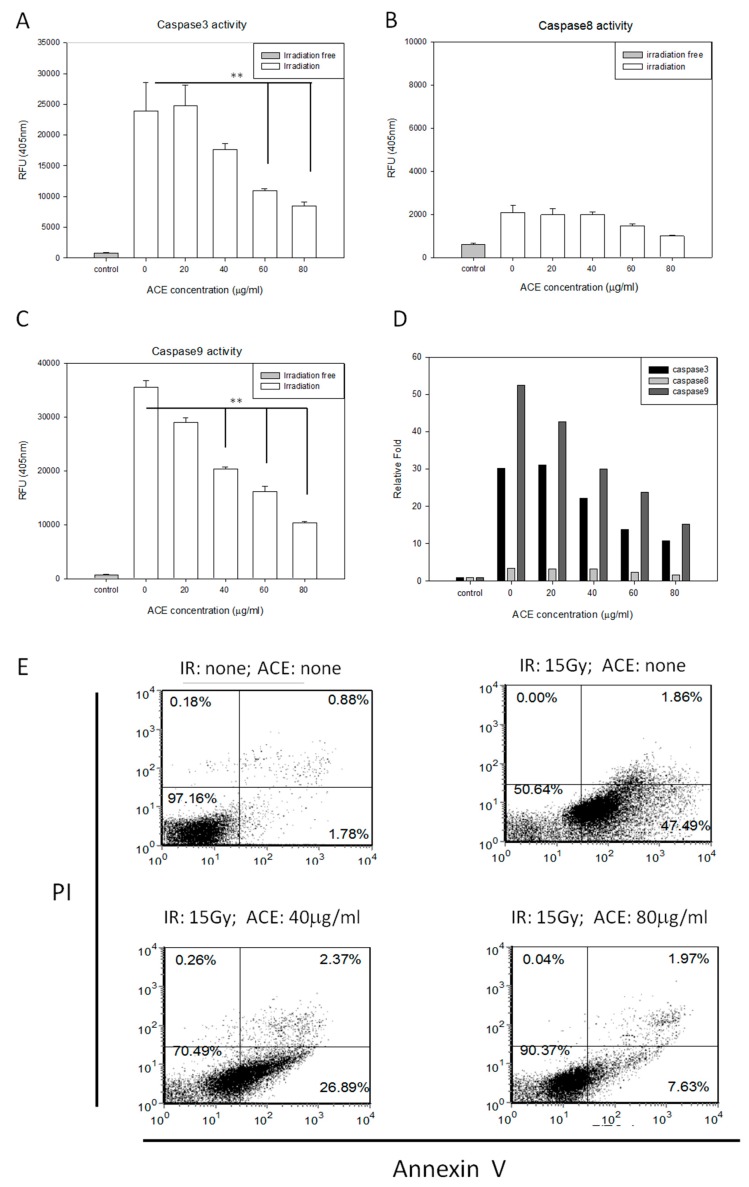
Protective effects of ACE against irradiation-induced apoptosis. CL48 cells were pre-treated with various concentrations of ACE for 16 h followed by irradiation at 15 Gy. Cells were harvested 48 h later for determination of caspase-3 activity (**A**), caspase-8 activity (**B**), and caspase-9 activity (**C**). The fold differences of the three caspase activities relative to individual control groups were also presented (**D**). For apoptosis analysis, cells were harvested 30 h after treatment followed by staining with PI and Annexin V (**E**). Results are representative of three independent experiments. The statistical analysis of the percentages of early plus late apoptotic cells (**F**). *, *p* < 0.05; **, *p* < 0.01, as compared with the irradiated ACE 0 μg/mL control group.

**Figure 3 ijms-20-00846-f003:**
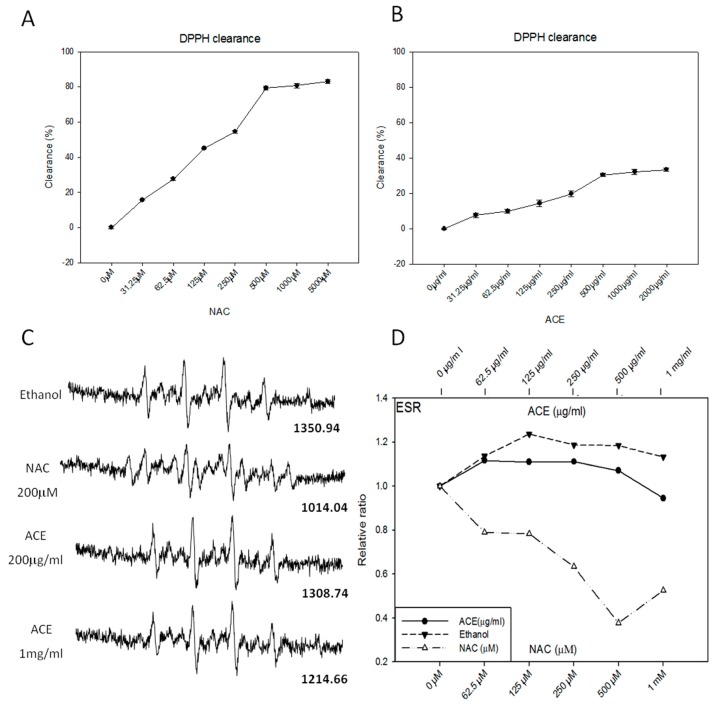
Examination of direct free radical-scavenging activities of ACE. DPPH free radical-scavenging activity of *N*-acetyl-l-cysteine (NAC) (**A**) and ACE (**B**) were analyzed. Results were obtained from three independent experiments, each experiment was done in triplicate. The ESR spectra of 5,5-dimethyl-1-pyrroline-N-oxide (DMPO) in phosphate buffered saline (PBS) were measured in the presence or absence of ACE or (*N*-acetyl-l-cysteine) (NAC) (200 μM) (**C**) and plotted and compared in a dose-dependent manner (**D**).

**Figure 4 ijms-20-00846-f004:**
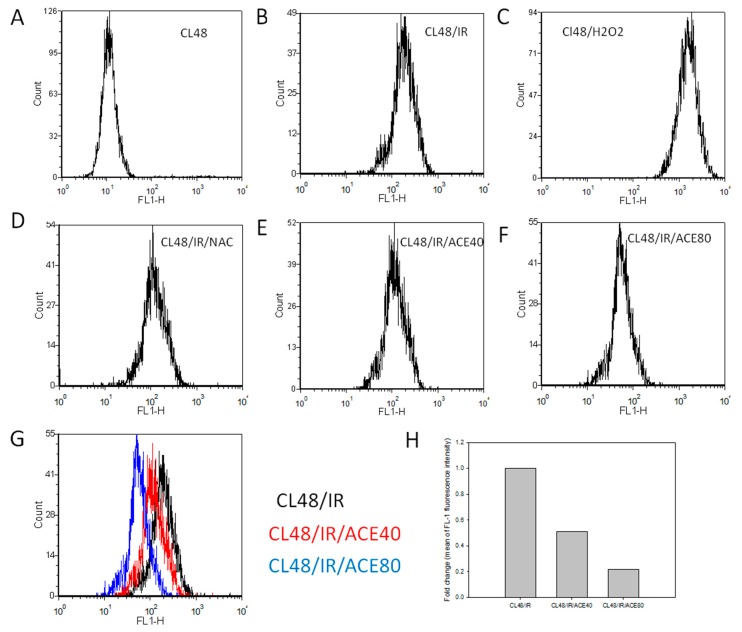
Reactive oxygen species (ROS) scavenging activity of ACE in CL48 cells. After being treated and harvested as described in the Materials and Methods section, cells were analyzed by flowcytometry after incubation with 2′,7′-dichlorofluorescin diacetate (DCFH-DA) for 15 min. (**A**) Control cells without treatment, (**B**) IR treated only, (**C**) H_2_O_2_ treated only, (**D**) pre-treated with NAC before IR, (**E**) pre-treated with ACE 40 μg/mL before IR, (**F**) pre-treated with ACE 80 μg/mL before IR. (**G**) Histograms of B, E, and F were combined to show the dose-dependent ROS scavenging activity of ACE. (**H**) Fold change of control fluorescence intensity by 40 μg/mL and 80 μg/mL of ACE, respectively.

**Figure 5 ijms-20-00846-f005:**
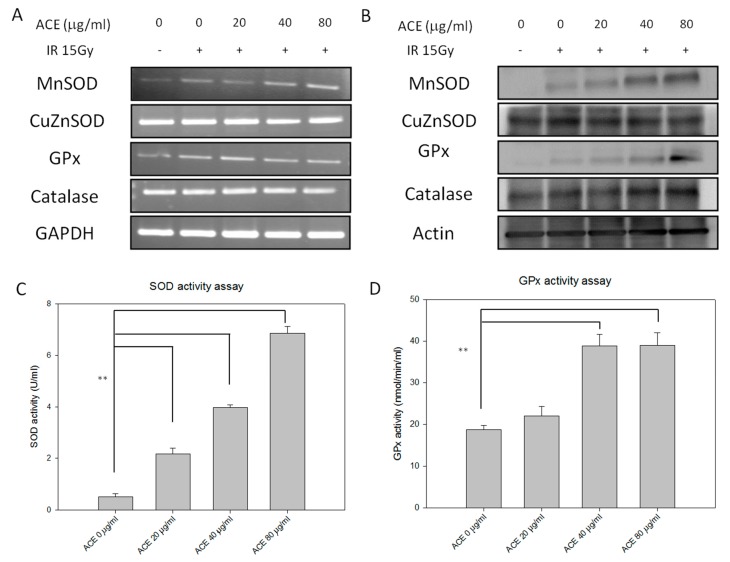
Redox-related enzymes expression and activity profiles of CL48 hepatocytes after IR and/or ACE treatments. With or without ACE pre-treatment for 16 h, cells were irradiated with a dose of 15 Gy and harvested 24 h later for RT-PCR (**A**), Western blot (**B**), total superoxide dismutase (SOD) activity (**C**), and glutathione peroxidase (GPx) activity (**D**) analyses. Results were obtained from three independent experiments, each experiment was done in triplicate. **, *p* < 0.01, as compared with the control IR only group.

**Figure 6 ijms-20-00846-f006:**
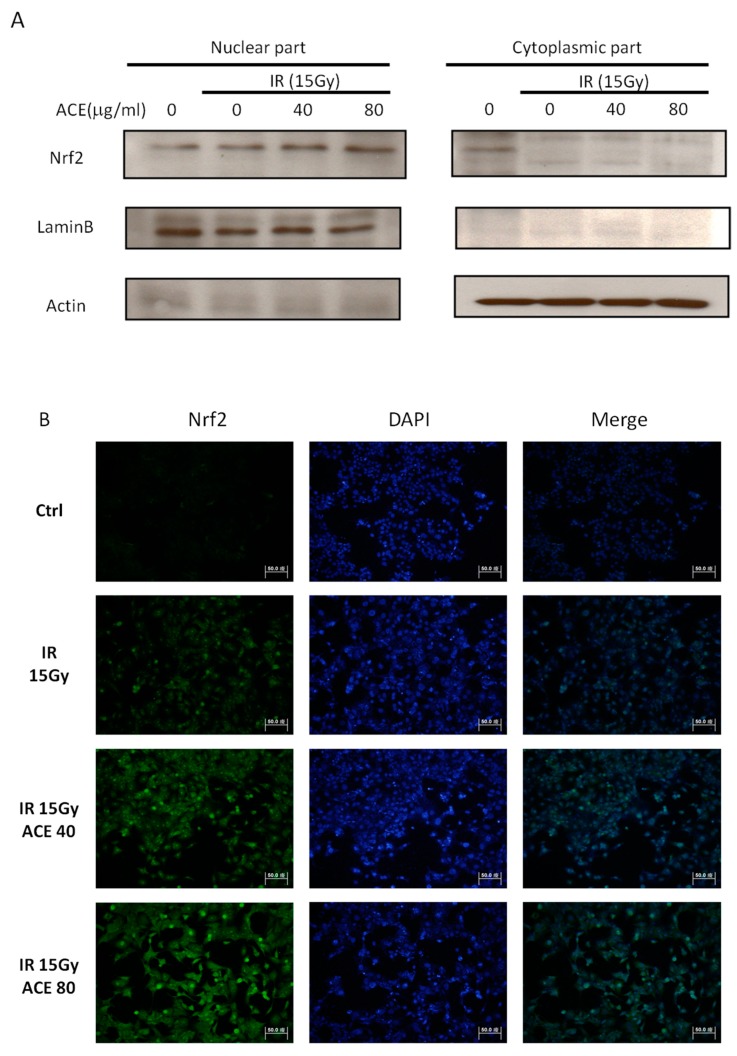
Enhancement of the irradiation-induced nuclear factor erythroid-2-related factor (Nrf2) expression and nuclear translocation by ACE treatment. With or without ACE pre-treatment for 16 h, CL48 cells were irradiated with a dose of 15 Gy and harvested 24 h later for Western blot (**A**) and immunofluorescence staining (**B**, left) of Nrf2. The slides were counterstained with 4′,6-diamidino-2-phenylindole (DAPI) (300 nM in PBS) (**B**, middle) and merged images (**B**, right). (**C**) The nuclear Nrf2 positivity rates were plotted. Results were obtained from random fields of three independent experiments for each group. *, *p* < 0.05; **, *p* < 0.01, as compared with the IR only group.

**Figure 7 ijms-20-00846-f007:**
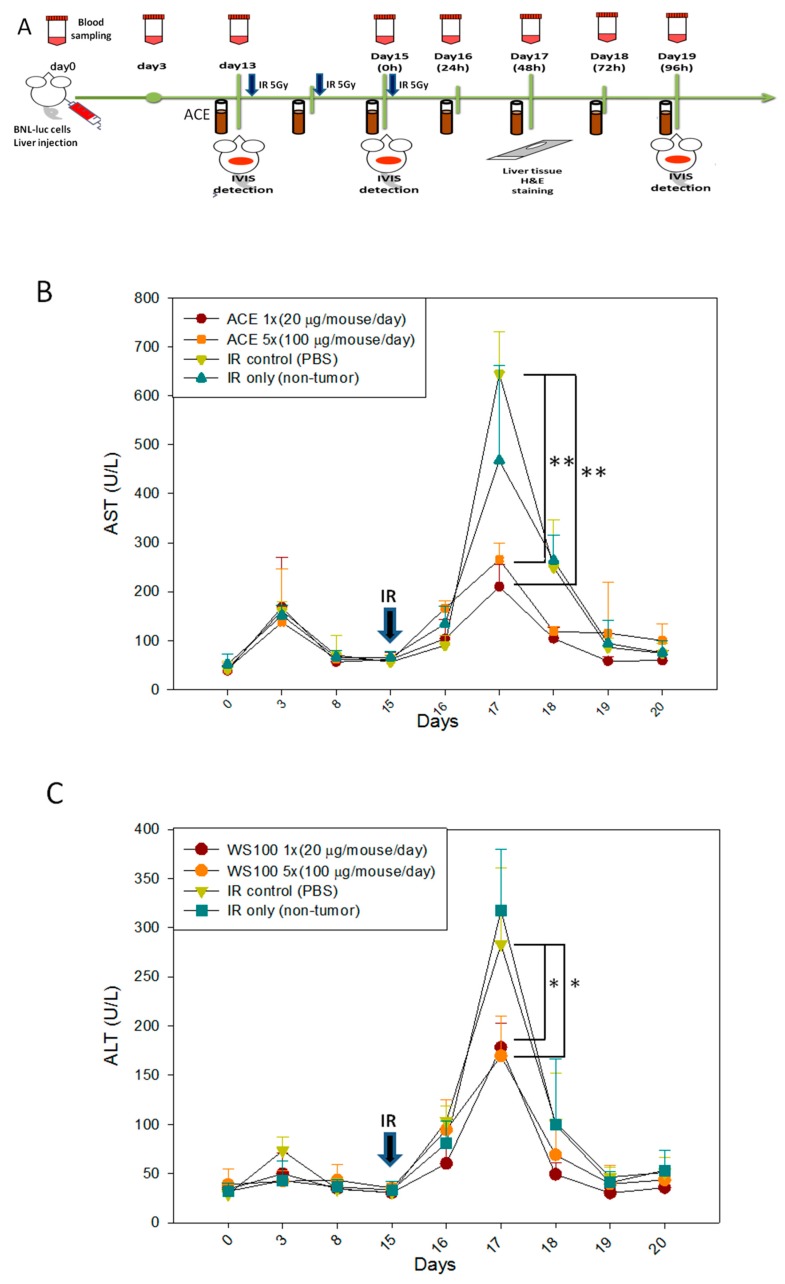
ACE alleviated the acute hepatitis markers in irradiated induces acute liver damage. (**A**) Scheme of the orthotropic hepatoma bearing mice model experiments. 48 h after irradiation, serum samples were collected every 24 h for monitoring the hepatitis markers aspartate transaminase (AST) (**B**) and alanine transaminase (ALT) (**C**). Liver tissues were harvested 96 h, upon sacrifice, and haematoxylin and eosin (H&E) stained for morphological detection of hepatic inflammation extent (the areas within black lines and marked with asterisks) (**D**) and Immunohistochemical (IHC) stained for Nrf2 detection (**E**). (**F**) The nuclear Nrf2 positivity rates were plotted. Results were obtained from random fields of three mice liver tissue sections for each group. *, *p* < 0.05; **, *p* < 0.01, as compared with the IR15Gy group.

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
