# Peer review of "Protective Effect of Antrodia cinnamomea Extract against Irradiation-Induced Acute Hepatitis"

_ijms, 2019, doi:10.3390/ijms20040846_

Round 1
Reviewer 1 Report
In the present manuscript, the authors examined the role of ACE against radiotoxicity both in normal liver cell line and in tumor-bearing mice. The authors demonstrated that ACE can protect animals from irradiation-induced liver damage by up-regulating the redox system during irradiation treatment. Although the present study is potentially easily understandable and very interesting, there are several issues that should be addressed as noted below.
1, How does ACE increase nuclear localized Nrf2 in the present study? The authors should discuss this point,
2, Please indicate throughout all figures the number of independent biological samples (n) and replicates. In Fig. 1, 2, 5, 6 and 7, the authors compared more than 2 groups, therefore the statistical significance must be determined by ANOVA, not using t-test. Please provide a new evaluation of these data.
Author Response
Reviewer #1
Comments and Suggestions for Authors
In the present manuscript, the authors examined the role of ACE against radiotoxicity both in normal liver cell line and in tumor-bearing mice. The authors demonstrated that ACE can protect animals from irradiation-induced liver damage by up-regulating the redox system during irradiation treatment. Although the present study is potentially easily understandable and very interesting, there are several issues that should be addressed as noted below.
1, How does ACE increase nuclear localized Nrf2 in the present study? The authors should discuss this point,
2, Please indicate throughout all figures the number of independent biological samples (n) and replicates. In Fig. 1, 2, 5, 6 and 7, the authors compared more than 2 groups, therefore the statistical significance must be determined by ANOVA, not using t-test. Please provide a new evaluation of these data.
Our responses to Reviewer 1
1. How does ACE increase nuclear localized Nrf2 in the present study? The authors should discuss this point.
Response:
Thanks for the comments. Since Nrf2 is the master regulator of cellular redox system and is involved in inflammatory response, we have now discussed several phytochemicals described in the literature that are capable of eliciting Nrf2 activation in liver cells, along with the underlying mechanisms. In addition, evidence in the literature shows that AC may induce the dissociation of Keap1 from Nrf2 and promote Nrf2 nuclear localization in immune cells. In our current study, we speculate that the ACE-increased nuclear translocation and activation of Nrf2 may also be achieved through this mechanism. How and what molecule of ACE achieves this effect remains to be investigated. We have now discussed these Nrf2 issues both in INTRODUCTION (page 2, lines 75-78) and DISCUSSION (page 14, lines 227-230 and lines 242-244).
2. Please indicate throughout all figures the number of independent biological samples (n) and replicates. In Fig. 1, 2, 5, 6 and 7, the authors compared more than 2 groups, therefore the statistical significance must be determined by ANOVA, not using t-test. Please provide a new evaluation of these data.
Response:
In general, each experiment was performed independently at least three times, each in triplicates. This information has been added in the text where appropriate, i.e., line 104 (page 3), line 123 (page 5), lines 135-136 (page 6), lines 306-307 and 350-351 (page 17), and lines 400-401 (page 19). For the animal model experiments, mice were divided into four groups and each group had six mice (page 12, lines 188-189); for the IHC staining, the samples were from three mice for each group (page 14, lines 206-207).
The data have been re-analyzed by using ANOVA (one-way analysis of variance), followed by post-hoc Tukey multiple comparison tests with significance level set at p<0.05. Statistical analysis was performed using SPSS version 20.0 (SPSS Inc., Chicago, IL) (Page 19, lines 400-403). We thank the reviewer for the kind and thoughtful comments. Fortunately, the conclusions drawn from our previous analyses hold after analysis by using ANOVA. To thank our biostatistician colleague Dr. Hsiu-Ying Ku for providing statistical assistance, we have added a statement on this regard to the acknowledgement section (page 19, lines 418-419).
Reviewer 2 Report
In their manuscript, Tsu-Hsiang Kuo et al. described a study about protective effects of Antrodia cinnamomea extract on irradiation-induced liver injury. They also observed an activation of NRF2 induced by Antrodia cinnamomea extract. The study is interesting, and can potentially expand the knowledge of Antrodia cinnamomea. There are still some minor issues need to be addressed.
There are some mistypings in the title of the manuscript.
Line 58. what does Chang&Chou mean?
There seems to be an extra sentence under Fig 1.
Fig.1. Why did author use two different ways to measure cytotoxicity of IR and ACE respectively?
Fig.2E. The authors need to analyze and performed statistical analysis for their flowcytometric analysis. The data can be shown as percentage of different cellular conditions.
Fig.4. Same as apoptosis analysis, DCFH-DA staining also need to be further analyzed. The data can be shown as fold change of control fluorescence intensity.
Author Response
Reviewer 2
Comments and Suggestions for Authors
In their manuscript, Tsu-Hsiang Kuo et al. described a study about protective effects of Antrodia cinnamomea extract on irradiation-induced liver injury. They also observed an activation of NRF2 induced by Antrodia cinnamomea extract. The study is interesting, and can potentially expand the knowledge of Antrodia cinnamomea. There are still some minor issues need to be addressed.
There are some mistypings in the title of the manuscript.
Line 58. what does Chang&Chou mean?
There seems to be an extra sentence under Fig 1.
Fig.1. Why did author use two different ways to measure cytotoxicity of IR and ACE respectively?
Fig.2E. The authors need to analyze and performed statistical analysis for their flowcytometric analysis. The data can be shown as percentage of different cellular conditions.
Fig.4. Same as apoptosis analysis, DCFH-DA staining also need to be further analyzed. The data can be shown as fold change of control fluorescence intensity.
Reviewer 2:
1. There are some mistypings in the title of the manuscript.
Response:
The two typos have been corrected. Thank you.
2. Line 58. what does Chang&Chou mean?
Response:
This is to indicate the nomenclature of Antrodia cinnamomea, which was first created by Chang and Chou and is now well recognized. To further clarify the expression and to avoid misunderstanding, we have now changed the sentence in line 58-59 (page 2) to “Antrodia cinnamomea (Chang & Chou) [10] is a valuable fungus originally found only in Taiwan” by deleting the abbreviation AC and adding a citation relating to its nomenclature.
3. There seems to be an extra sentence under Fig 1.
Response:
The extra sentence is deleted. Thank you.
4. Fig.1. Why did author use two different ways to measure cytotoxicity of IR and ACE respectively?
Response:
In our experience, the two methods always give rise to same results in estimating cell numbers. Since the cell counting method is much more tedious and time consuming, once we repeatedly confirmed the equality of the two methods in our study, we then used only the MTT assay for all of the following experiments in evaluating cell numbers.
5. Fig.2E. The authors need to analyze and performed statistical analysis for their flowcytometric analysis. The data can be shown as percentage of different cellular conditions.
Response:
We have performed one-way ANOVA test to analyze the flowcytometric data of Fig 2E. Percentages of the early and late apoptotic cells are combined and compared statistically between groups. To clearly describe the effects of ACE on irradiation-induced apoptosis based on this analysis, we have added the following sentence to the corresponding text (page 3, lines 110-113): “Percentages of the early and late apoptotic cells were combined and compared statistically. As shown, apoptosis cells decreased from 52% to 34% and 14 by ACE 40 μg/ml and 80 μg/ml, respectively (Fig 2F)”.
6. Fig.4. Same as apoptosis analysis, DCFH-DA staining also need to be further analyzed. The data can be shown as fold change of control fluorescence intensity.
Response:
To indicate the effects of ACE on irradiation-induced ROS production, we have adopted the “fold change of control” method as suggested by the reviewer and the following sentence is appended to the legend (page 8, lines 150-151): “(H) Fold change of control fluorescence intensity by 40μg/ml and 80μg/ml of ACE, respectively.” The corresponding sentence is also added to the RESULT section (page 6, lines 140-142): “ROS levels were decreased 50% and 78% by ACE 40 μg/ml and 80 μg/ml, respectively (Fig. 4G and 4H).”
Round 2
Reviewer 1 Report
The authors replied to my questions and comments well. I have no comment.